# Small and sick newborn care: Changes in service readiness scoring between baseline and 2023 for 65 neonatal units implementing with NEST360 in Kenya, Malawi, Nigeria, and Tanzania

Rebecca E. Penzias[1]*, Morris Ondieki Ogero[1], Robert Tillya[2], Irabi Kassim[2], Olabisi Dosunmu[3], Opeyemi Odedere[3,4], Hannah Mwaniki[5], Vincent O. Ochieng[5], Dolphine Mochache[5], Samuel K. Ngwala[6], Evelyn Zimba[4], Grace T. Soko[6], Christine Bohne[2,4], David Gathara[1], James H. Cross[1], Josephine Shabani[1,2], Catherine Paul[2,4], Donat Shamba[1,2], Honorati Masanja[2], Nahya Salim[2,7], Charles Osuagwu[3], Afeez Idowu[3], Ifeanyichukwu Anthony Ogueji[3], Olukemi Tongo[8], Olabanjo Okunlola Ogunsola[3], Veronica Chinyere Ezeaka[9], Ekran Rashid[5], George Okello[4], John Wainaina[10], William M. Macharia[5], Msandeni Chiume[6,11], Alfred Chalira[12], Queen Dube[6,12], Edith Gicheha[4], Elizabeth M. Molyneux[6], Millicent Alooh[4], Simon Cousens[1], Maria Oden[4], Rebecca Richards-Kortum[4], Joy E. Lawn[1‡], Eric O. Ohuma[1‡]

1 Centre for Maternal, Adolescent, Reproductive, & Child Health, London School of Hygiene & Tropical Medicine, London, United Kingdom, 2 Ifakara Health Institute, Ifakara, Tanzania, 3 APIN Public Health Initiatives, Abuja, Nigeria, 4 Rice360 Institute for Global Health Technologies, Rice University, Houston, Texas, United States of America, 5 Department of Paediatrics, Aga Khan University, Nairobi, Kenya, 6 Kamuzu University of Health Sciences (formerly College of Medicine, University of Malawi), Blantyre, Malawi, 7 Department of Paediatrics and Child Health, Muhimbili University of Health and Allied Sciences, Dar Es Salaam, Tanzania, 8 Department of Paediatrics, College of Medicine, University of Ibadan, Ibadan, Nigeria, 9 Department of Paediatrics, College of Medicine, University of Lagos, Lagos, Nigeria, 10 Kenya Medical Research Institute (KEMRI)-Wellcome Trust, Nairobi, Kenya, 11 Kamuzu Central Hospital, Lilongwe, Malawi, 12 Ministry of Health, Lilongwe, Malawi

‡ Joint senior authors.
* rebecca.penzias@lshtm.ac.uk

## Abstract

Health Facility Assessments (HFAs) are important for measuring and tracking service readiness for small and sick newborn care (SSNC). NEST360 Alliance aims to reduce neonatal mortality in four countries (Kenya, Malawi, Nigeria, Tanzania). NEST360 and UNICEF facilitated HFA tool design with ministries of health in four African countries and developed two complimentary approaches to summarise readiness. Using the NEST360/UNICEF HFA tool, we collected data, developed two service readiness scoring approaches for SSNC (standards-based scoring by adapted World Health Organization (WHO) health system building blocks (HSBBs) and assessing service readiness across the health system, and level-2+ scoring by WHO clinical interventions), and applied across 65 neonatal units implementing NEST360. Service readiness change was assessed between baseline (Sept 2019-March 2021) and follow-up HFA (May-July 2023). For each neonatal unit, a percentage difference

**Data availability statement:** All partners in the NEST360 alliance collaborated to create and sign data sharing and transfer agreements. The minimal dataset generated during the current study have been deposited online with data access subject to approval at https://doi.org/10.17037/data.00004609.

**Funding:** This work is funded through the NEST360 alliance with thanks to John D. and Catherine T. MacArthur Foundation, the Bill & Melinda Gates Foundation, ELMA Philanthropies, The Children's Investment Fund Foundation UK, The Lemelson Foundation, The Sall Family Foundation, and the Ting Tsung and Wei Fong Chao Foundation under agreements to William Marsh Rice University as the prime grant holder under sub agreement to LSHTM. The funders had no role in the study design, data collection and analysis, decision to publish, or manuscript preparation.

**Competing interests:** The authors have declared that no competing interests exist.

score was computed between baseline and 2023 HFA scores. Scores were calculated for each neonatal unit as the unit of analysis, and disaggregated by HSBB, clinical intervention, and sub-modules. Data from 65 neonatal units were analysed, i.e., 36 in Malawi, 13 in Kenya, 7 in Tanzania, and 9 in Nigeria. Median time between baseline and 2023 HFAs was 31 months [IQR 29–34 months]. Median baseline and 2023 scores were 41% [IQR 35–52%] and 55% [IQR 46–62%] respectively with 14% median score change [IQR 4–18%] for level-2+ scores. For standards-based scores, median baseline and 2023 scores were 51% [IQR 48–58%] and 60% [IQR 54–66%] respectively with a 9% median score change [IQR 3–11%]. Hospitals in Tanzania [Median 24%, IQR 16–30%] and Nigeria [Median 28%, IQR 17–30%] showed greater improvements on average for level-2+ scores compared to hospitals in Kenya and Tanzania. Data on changes in service readiness scores can be used to track service readiness over time, benchmark between hospitals, identify gaps, and assess progress towards newborn targets.

## Introduction

Sixty-four countries are off track for the global neonatal survival target for 2030, and as such, there is increasing focus on neonatal care [1]. The Every Newborn Action Plan set coverage targets including one for small and sick newborn care (SSNC); that 80% of districts in every country have at least one functional level-2 inpatient newborn care unit with respiratory support by 2025 [2]. Tracking progress towards this target requires repeated assessments of service readiness of level-2 inpatient newborn care units.

Health facility assessments (HFAs) are often used to measure service readiness, such as for emergency obstetric care and general service provision, and more recently have been used to assess readiness for SSNC [3–6]. A HFA tool for SSNC was co-designed by the NEST360 (Newborn Essential Solutions and Technologies) alliance together with the United Nations Children's Fund (UNICEF) in partnership with ministries of health in four African countries, and has since been used in more than 135 hospitals [7]. Two scoring approaches were developed to quantify HFA data for SSNC: a) standards-based, including items for SSNC service readiness by health system building block (HSBB), and scored on availability and functionality, and b) level-2+, scoring items on readiness to provide WHO level-2+ clinical interventions [8].

Quantifying service readiness can support hospitals in identifying health systems gaps in readiness to provide care, so that these gaps may be addressed to improve quality of care generally or for specific clinical interventions. However, few studies have tracked comparable HFA scores over time. It is also important for policy makers and implementers to understand what is driving changes in service readiness, and where there may be continuing gaps in SSNC readiness. These assessments can support identification of additional improvements and support progress towards global newborn targets.

A number of studies have assessed readiness for basic or comprehensive emergency obstetric care, and included some assessment of essential newborn care interventions [9–11], but few have assessed service readiness for neonatal care, and even fewer have published repeated measurements of newborn care service readiness [12–14]. Of the studies with repeated measurements for newborn care, we found none that assessed changes in readiness for level-2+ SSNC, noting that a tool assessing level-2+ SSNC was not available until recently [7].

## Aim

We aim to evaluate the changes in SSNC service readiness using standards-based health system building blocks and level-2+ clinical intervention scores over time in 65 neonatal units implementing with NEST360, a programme aimed at reducing inpatient neonatal mortality, in Kenya, Malawi, Nigeria, and Tanzania. Specifically, we aim to; a) evaluate changes in standards-based and level-2+ service readiness between baseline (2019–2021) and 2023 follow-up scores for each neonatal unit according to Health System Building Block (HSBB) or clinical intervention, and b) evaluate changes in standards-based and level-2+ service readiness for each neonatal unit according to HSBB or clinical intervention disaggregated by sub-module scores.

## Methods

### NEST360

The NEST360 Alliance, which started in 2019, is working with ministries of health in Kenya, Malawi, Nigeria, and Tanzania to reduce newborn deaths in hospitals by implementing a government-led health systems package. NEST360 particularly focuses on providing a bundle of innovative technologies with training and mentoring for clinicians and biomedical technicians, strengthening information systems, and implementing other evidence-based strategies for high-quality and sustainable newborn care, including government-led continuous quality improvement (QI) [15,16]. Quarterly QI visits began in 2021 with an average of three visits per year at each hospital. The NEST360 Alliance has supported implementation in 67 neonatal units in Kenya (n = 13), Malawi (n = 38), Nigeria (n = 9 at 7 hospitals), and Tanzania (n = 7).

### Data sources and data collection

**Health facility assessment tool.** A HFA tool for SSNC was designed by NEST360 and UNICEF in partnership with national governments of four African countries, and systematically developed using a three-step evidence-based process to assess readiness to provide SSNC [7]. The tool includes 1508 items across ten discrete modules and is organised into adapted HSBBs: infrastructure, medical devices and supplies, human resources, information systems, family centred care, and governance. The tool was used to collect cross-sectional data on hospital readiness prior to NEST360 implementation at 67 neonatal units implementing NEST360 in Kenya, Malawi, Nigeria, and Tanzania during the period September 2019 - March 2021 (HFA – Baseline) (S1 File) [17]. The tool was used to collect follow-up data at 65 neonatal units from May to July 2023 (HFA – 2023), as two hospitals in Malawi did not receive follow-up HFAs due to delayed hospital opening and implementation timelines. HFA data collection was completed by a team of seven multidisciplinary assessors in one day at each hospital using observation and interviewer-led assessment with hospital staff, including nurses, biomedical technicians, laboratory managers, hospital management, and other relevant department staff, and collecting data on a mobile REDCap application on Android tablets [18]. Data collectors were trained for four days before data collection began. Data were checked by HFA team supervisors and the national database manager. All HFA data were uploaded to and stored on servers of the designated country partner during and after data collection. De-identified pooled HFA data were transferred to a central database for analysis.

**Health facility assessment scoring: Standards-based and level-2+.** Two summary scoring approaches were developed to quantify service readiness for SSNC: standards-based, and level-2+ scoring [8]. Standards-based

scores were organised by the HSBB framework adapted from WHO, and include all items required for SSNC service readiness, and were scored on availability and functionality [17]. Within each HSBB, items were organised into HSBB sub-modules according to content area (i.e., education within the human resources HSBB). Level-2+ scores were organised by the ten WHO level-2+ clinical interventions, and were scored based on readiness to provide basic, comprehensive, or gold-standard care for that intervention [2]. Within each clinical intervention, items were organised into intervention sub-modules to include all potential pathways for diagnosis/screening and treatment/management for that intervention (i.e., cup feeding and nasogastric tube feeding within the assisted feeding intervention). Standards-based scores focus on a health systems approach and assess service readiness across the health system, and include 1043 (69%) of the 1508 HFA items. The level-2 + scores focus specifically on interventions provided on the neonatal unit, and as such, include fewer items with 309 (20%) of 1508 items scored (S2 File). Both scores can be useful for identifying gaps in readiness to provide care, though it may be more feasible to track level-2+ scores over time as fewer items are included in the scores. Scores were aggregated and summarised as a percentage of the maximum possible score and equally weighted for each standards-based and level-2+ intervention (excluding follow-up of at-risk newborns) to obtain an overall score by neonatal unit, HSBB and HSBB sub-module, and intervention and intervention sub-module.

The neonatal unit was the unit of analysis, and two hospitals in Nigeria had different scores for inborn and outborn neonatal units. Hospitals were categorised into primary, secondary, and tertiary hospitals according to the level of care expected in each country. Items that were only assessed at baseline or 2023 HFAs were excluded from score change analyses. Fewer than 5% of items in each category were excluded from score change analyses. Items with missing responses were removed from the denominator of scores [8].

## Data analyses

To calculate change in HFA baseline and 2023 scores, absolute percentage point score difference was computed at the individual neonatal unit level.

Scores and score sub-types were aggregated by computing percent change in median and interquartile ranges for absolute percent change and are reported disaggregated by HSBB and clinical intervention, by sub-modules and by country. We also chose a priori to disaggregate scores by hospital-level (tertiary, secondary, and primary) and annual number of admissions. After exploration, we decided to report scores disaggregated by hospital-level and admissions for Malawi only given the large number of hospitals in Malawi and implementation in all districts throughout the country. Secondary and tertiary level hospitals, and high-volume primary hospitals were analysed separately from low-volume primary hospitals. High-volume primary hospitals included all primary hospitals with admissions per year exceeding the 75% percentile of hospital admissions across Malawi.

Relative gap reduction was also calculated to determine the proportion of the service readiness gap closed between baseline and 2023 follow up. As hospitals should have all items to be ready for service provision, the gap was assumed to be the difference between a score of 100% and the baseline score.

Scoring changes were examined visually using heatmaps, boxplots, and bar graphs, and by comparing differences in absolute score percent difference and relative gap reduction and score ranges to assess dispersion. Heatmaps were computed using the median of the baseline score, percent change in pooled median absolute percent difference, and interquartile range of median absolute percent difference by neonatal unit. Overall standards-based and level-2+ score changes were also analysed using quantile regression to adjust for the different baseline scores within and between countries. Heatmaps present absolute percentage difference in median scores using five colours to represent each 10% change in scores.

HFA data were cleaned, and quality checked in REDCap [18]. Visualisations were generated using Stata 17 and Microsoft Excel (2023). All analyses were performed in Stata 17 (StataCorp LLC, Texas, USA).

**Ethical approval and inclusivity in global research**

Ethical approval was received from a local institutional review board in each country and the London School of Hygiene & Tropical Medicine ethics committee (no. 21892) (S3 File). The NEST360 Alliance data sharing agreement covered data sharing between organisations. Individual consent was not required because no personal identification data were included. Additional information regarding the ethical, cultural, and scientific considerations specific to inclusivity in global research is included in the Supporting Information (S1 Checklist).

## Results

Data from 65 neonatal units were analysed (36 in Malawi, 13 in Kenya, 7 in Tanzania, and 9 in Nigeria). Of the 65, 35 neonatal units were at secondary and tertiary hospitals, and 30 were at primary hospitals, noting that all primary hospitals were in Malawi. Of the 36 hospitals in Malawi, median annual admissions to the neonatal unit was 963 (IQR 609–1338). Of the 30 primary hospitals in Malawi, 5 were high-volume hospitals with admissions above the 75% percentile (S4 File). Median time between baseline and 2023 HFAs was 31 months [IQR 29–34 months]. Median time between baseline HFA and start of implementation was 2 months [IQR 1–5 months].

**Objective 1: To evaluate change between baseline and 2023 in standards-based and level-2+ service readiness scores and by HSBB or clinical intervention, overall and by country**

**Percent change overall and by country.** There was a wider dispersion in level-2+ score changes compared to standards-based scores for all countries (Fig 1). For all hospitals, the median baseline score was 41% [IQR 35–52%] for level-2+ scores and 51% [IQR 48–58%] for standards-based scores (Fig 1). The median 2023 score was 55% [IQR 46–62%] for level-2+ scores and 60% [IQR 54–66%] for standards-based scores.

For all hospitals, the change in median absolute percentage was 14% [IQR 4–18%] for level-2+ scores and 9% [IQR 3–11%] for standards-based scores. Hospitals in Tanzania [Median 24%, IQR 16–30%] and Nigeria [Median 28%, IQR 17–30%] showed greater improvements on average for level-2+ scores compared to hospitals in Malawi [Median 8%, IQR -2–12%] and Kenya [Median 7%, IQR 2–15%] (Fig 1). Hospitals in Tanzania [Median 15%, IQR 8–15%] and Nigeria [Median 13%, IQR 6–13%] showed greater increase in standards-based scores compared to hospitals in Malawi [Median 6%, IQR 3–9%], and Kenya [Median 4%, IQR 3–10%]. Using Tanzania as a baseline, after adjusting for baseline scores, hospitals in Malawi had 8% lower standards-based scores on average compared to hospitals in Tanzania (p = 0.01), and hospitals in Kenya and Nigeria had 4% (p = 0.19) and 2% (p = 0.45) lower standards-based scores respectively. Similarly, after adjusting for baseline scores, hospitals in Malawi and Kenya had 16% (p = 0.001) and 12% (p = 0.03) lower level-2+ scores respectively compared to hospitals in Tanzania. In Malawi, low-volume primary hospitals (<1300 annual admissions) [Median 10%, IQR 5–14%] showed greater improvements in level-2+ scores compared to high-volume primary, secondary, and tertiary hospitals (>=1300 annual admissions) [Median 2%, IQR -10–12%]. Similar trends in overall and country-specific score changes were observed for relative score changes.

**Percent change by HSBB.** For standards-based scores, absolute percent change in HSBB scores in all hospitals was highest for human resources [Median 13%, IQR 5–18%], information systems [Median 11%, IQR 2–15%], and medical devices and supplies [Median 9%, IQR 5–13%], compared to family centred care [Median 7%, IQR -3–15%], infrastructure [Median 4%, IQR -4–7%], and governance and leadership [Median 1%, IQR -10–10%] (S5 File). Absolute percent change in HSBB scores was higher on average in hospitals in Nigeria [Median 13%, IQR 6–13%] and Tanzania [Median 15%, IQR 8–15%] compared to hospitals in Malawi [Median 6%, IQR 3–9%] and Kenya [Median 4%, IQR 3–10%] (Fig 2). Notably, hospitals in Tanzania had higher percent change for family centred care [Median 21%, IQR 2–24%] compared to hospitals in Malawi [Median 8%, IQR 0–14%], Kenya [Median 3%, IQR

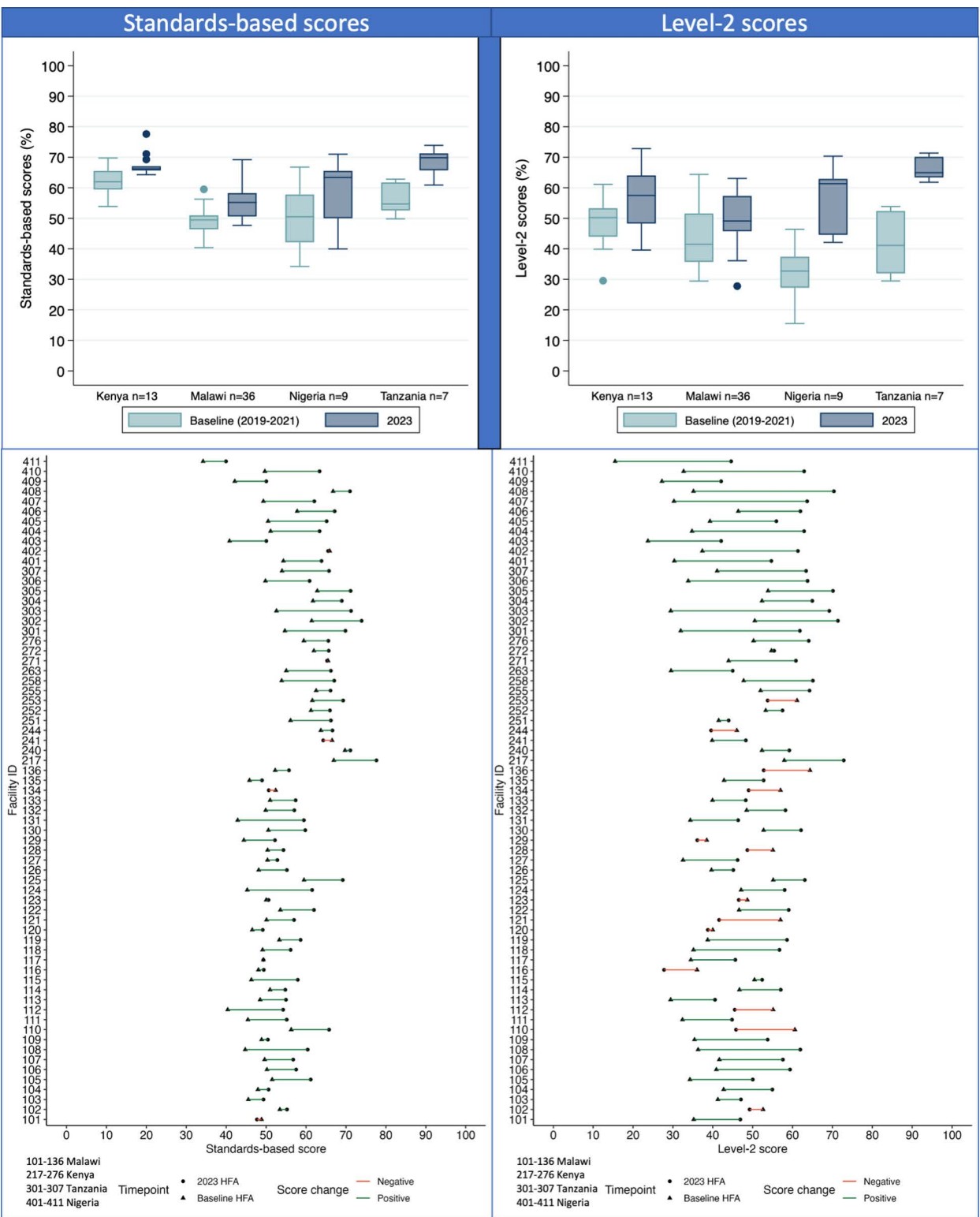

**Fig 1. Standards-based and level-2+ Health Facility Assessment (HFA) scores at baseline (2019-2021) and 2023 for 65 neonatal units implementing with NEST360.**

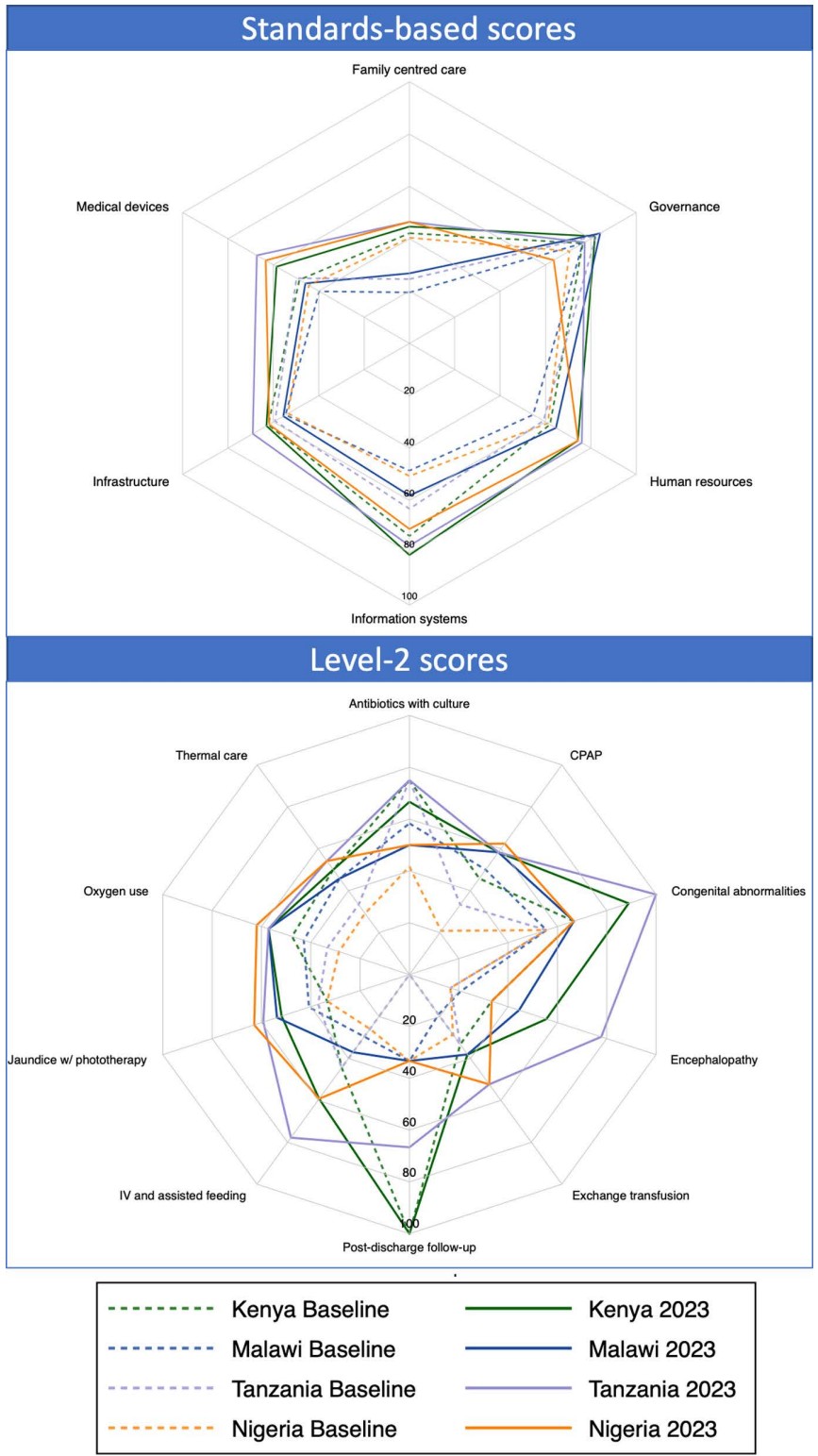

**Fig 2. Spider plot of median standards-based scores by health system building blocks and level-2+ scores by clinical intervention and country for 65 NEST360 implementing neonatal units at baseline (2019-2021) and 2023 follow-up.** Abbreviations: IV – Intravenous; CPAP – Continuous Positive Airway Pressure.

-7–15%], and Nigeria [Median 6%, IQR -4–16%]. High-volume hospitals also showed great improvements in family centred care readiness [Median 15%, IQR 10–19%]. Despite improvements in family centred care scores, these scores remained low at 2023 HFAs [Median 34%, IQR 23–46%]. Hospitals in Malawi showed the greatest increase in readiness for governance and leadership [Median 8%, IQR -8–20%], though this was the highest-scoring HSBB at baseline in most hospitals and was primarily driven by improvements in low-volume primary hospital governance scores [Median 5%, IQR −3–20%].

**Percent change by clinical intervention.** For level-2+ scores, the absolute percent change in median intervention scores was highest for detection and management of neonatal encephalopathy [Median 31%, IQR 0–40%], detection and management of jaundice with phototherapy [Median 19%, IQR 7–22%], safe administration of oxygen [Median 14%, IQR 5–24%] and management of exchange transfusion [Median 14%, IQR 5–24%], compared to thermal care [Median 4%, IQR -8–17%], referral for congenital abnormalities [Median 11%, IQR 0–22%], intravenous and assisted feeding [Median 11%, IQR 7–22%], and Continuous Positive Airway Pressure (CPAP) management [Median 12%, IQR 4–25%]. There was no consistent change in readiness for neonatal sepsis management [Median 0%, IQR -17–8%] and follow-up of at-risk newborns [Median 0%, IQR -33–33%]. Readiness increased for most hospitals across countries, however, there was a decrease in readiness for neonatal sepsis management in hospitals in Kenya [Median -8%, IQR -33–0%] and Malawi [Median -8%, IQR -17–8%], though this was one of the highest-scoring interventions at baseline for most hospitals, so had less room for improvement (Fig 2). Major improvements in readiness for CPAP management [Median 41%, IQR 29–46%] and safe oxygen use [Median 33%, IQR 24–38%] were seen in hospitals in Nigeria. In Tanzania, there were major improvements in hospital readiness for detection and management of neonatal encephalopathy [Median 61%, IQR 19–69%], and follow-up of at-risk newborns [Median 67%, IQR 0–100%]. In Malawi, low-volume primary hospitals showed greater improvements in detection and management of neonatal encephalopathy [Median 31%, IQR 11–39%] and detection and management of jaundice with phototherapy [Median 15%, IQR 4–19%] compared to high-volume primary, secondary, and tertiary hospitals [Encephalopathy: median −22%, IQR −31–28%; Jaundice: median 7%, IQR 4–15%].

**Objective 2: To evaluate change in standards-based and level-2+ service readiness sub-module scores for SSNC at baseline and 2023 follow-up overall and by country**

**Standards-based drivers of absolute percent change.** There were also notable changes in sub-modules for standards-based scores. Overall, increased readiness for medical devices and supplies was driven by improvements in biomedical workshop readiness [Median 27%, IQR 13–30%] and medical device requirement readiness [Median 12%, IQR 4–20%], which includes availability and functionality of key devices for small and sick newborn care (Fig 3). In Tanzania, improvements in this HSBB were also driven by increased pharmacy readiness [Median 14%, IQR 10–32%]. Overall improvements in human resource readiness were driven by increased education [Median 12%, IQR 4–24%] and enabling environment readiness [Median 12%, IQR 1–19%]. However, in Tanzania, this was driven primarily by improvements in enabling environment readiness [Median 25%, IQR 11–36%]. Overall, increased readiness for information systems was driven by foundations for information systems readiness [Median 16%, IQR 3–26%], which focuses on infrastructure needed to support hospital-based information systems. In Kenya, there was less change in information systems readiness [Median 7%, IQR 0–12%], though some improvements in foundations for information systems were noted [Median 11%, IQR 1–18%].

**Level-2+ drivers of absolute percent change.** Similar to standards-based scores, there were also notable changes in sub-module scores for level-2+ interventions. Overall, increased readiness for intravenous and assisted feeding was driven by improvements in intravenous (IV) fluid equipment and consumables [Median 67%, IQR 0–67%] and infection

| | n items | Total | | Malawi | | Kenya | | Tanzania | | Nigeria | |
|---|---|---|---|---|---|---|---|---|---|---|---|
| | | Baseline | % change (IQR) | Baseline | % change (IQR) | Baseline | % change (IQR) | Baseline | % change (IQR) | Baseline | % change (IQR) |
| **Overall** | 1043 | 51 | 9 (3 to 11) | 49 | 6 (3 to 9) | 62 | 4 (3 to 10) | 55 | 15 (8 to 15) | 50 | 13 (6 to 13) |
| **Medical devices and supplies** | 544 | 43 | 9 (5 to 13) | 40 | 6 (5 to 9) | 49 | 10 (7 to 11) | 50 | 17 (12 to 24) | 44 | 19 (13 to 23) |
| Medical device requirements | 192 | 35 | 12 (4 to 20) | 35 | 7 (4 to 13) | 42 | 9 (4 to 13) | 40 | 20 (11 to 26) | 33 | 26 (21 to 39) |
| Laboratory | 100 | 75 | -3 (-4 to 6) | 66 | 1 (-6 to 5) | 81 | 0 (-1 to 7) | 89 | 2 (0 to 5) | 81 | 4 (-2 to 12) |
| Pharmacy | 128 | 53 | 4 (-4 to 11) | 49 | -2 (-8 to 6) | 61 | 3 (1 to 12) | 65 | 14 (10 to 32) | 52 | 7 (0 to 15) |
| Biomedical workshop* | 124 | 12 | 27 (13 to 30) | 9 | 19 (13 to 24) | 23 | 24 (11 to 35) | 18 | 27 (19 to 37) | 16 | 27 (11 to 35) |
| **Human resources** | 229 | 57 | 13 (5 to 18) | 55 | 10 (2 to 15) | 62 | 12 (7 to 20) | 59 | 17 (10 to 24) | 61 | 13 (5 to 22) |
| People | 18 | 33 | 0 (-3 to 7) | 26 | 4 (-2 to 8) | 42 | -1 (-7 to 5) | 34 | 0 (-7 to 6) | 52 | -1 (-6 to 2) |
| Education | 150 | 78 | 12 (4 to 24) | 80 | 10 (5 to 24) | 77 | 14 (6 to 30) | 80 | 8 (3 to 14) | 71 | 18 (3 to 30) |
| Enabling Environment | 61 | 62 | 12 (1 to 19) | 61 | 7 (-1 to 13) | 69 | 9 (10 to 22) | 60 | 25 (11 to 36) | 65 | 18 (2 to 21) |
| **Infrastructure** | 131 | 56 | 4 (-4 to 7) | 55 | 1 (-5 to 6) | 62 | 1 (-2 to 3) | 59 | 10 (0 to 14) | 54 | 8 (4 to 12) |
| Electrical power | 25 | 58 | 3 (-2 to 11) | 59 | 1 (-6 to 7) | 53 | 5 (-2 to 15) | 57 | 4 (-2 to 11) | 58 | 11 (5 to 14) |
| Medical gases and vacuum | 5 | 12 | -2 (-5 to 10) | 0 | 0 (0 to 0) | 75 | -25 (-40 to 7) | 12 | 38 (0 to 35) | 50 | 0 (0 to 25) |
| Referral | 11 | 51 | 5 (-8 to 12) | 49 | 0 (-9 to 8) | 65 | -5 (-5 to 2) | 49 | 21 (-7 to 30) | 44 | 16 (-2 to 33) |
| Space and design | 36 | 48 | 5 (-3 to 13) | 41 | 10 (2 to 15) | 58 | 2 (-5 to 11) | 61 | 13 (6 to 29) | 52 | 1 (-12 to 9) |
| Water, sanitation, and hygiene | 54 | 67 | -1 (-8 to 11) | 67 | -5 (-16 to 10) | 67 | 3 (-8 to 4) | 75 | 0 (-1 to 7) | 58 | 8 (3 to 15) |
| **Information Systems** | 95 | 54 | 11 (2 to 15) | 49 | 9 (3 to 14) | 74 | 7 (0 to 12) | 63 | 14 (13 to 19) | 51 | 20 (2 to 27) |
| Data collection | 59 | 65 | 7 (-1 to 10) | 64 | 5 (3 to 9) | 83 | 5 (-2 to 11) | 61 | 12 (1 to 15) | 65 | 7 (-1 to 7) |
| Data management | 6 | 59 | 8 (-6 to 20) | 54 | 7 (-4 to 16) | 88 | -5 (-6 to 7) | 35 | 43 (19 to 48) | 65 | -4 (-31 to 20) |
| MPDSR* | 5 | 50 | 0 (-17 to 33) | 17 | 25 (-17 to 33) | 67 | 0 (-50 to 17) | 83 | 17 (0 to 17) | 67 | -17 (-50 to 17) |
| Foundations | 25 | 41 | 16 (3 to 26) | 36 | 11 (3 to 19) | 64 | 11 (1 to 18) | 66 | 19 (11 to 29) | 25 | 49 (-1 to 60) |
| **Family Centred Care** | 30 | 27 | 7 (-3 to 15) | 19 | 8 (0 to 14) | 42 | 3 (-7 to 15) | 25 | 21 (2 to 24) | 40 | 6 (-4 to 16) |
| Organisation of care | 25 | 25 | 7 (-4 to 17) | 17 | 6 (0 to 13) | 42 | 1 (-12 to 11) | 21 | 24 (1 to 28) | 44 | 1 (-3 to 18) |
| Discharge and early development* | 2 | 33 | 0 (0 to 33) | 33 | 0 (0 to 0) | 100 | 0 (0 to 67) | 33 | 67 (0 to 67) | 67 | 0 (-33 to 33) |
| Kangaroo Mother Care (KMC) | 2 | 25 | 25 (-25 to 25) | 38 | 12 (-25 to 25) | 33 | 17 (-25 to 17) | 50 | 25 (0 to 50) | 0 | 0 (0 to 0) |
| Parent power* | 1 | 50 | 0 (0 to 0) | 25 | 25 (0 to 25) | 50 | 0 (0 to 0) | 50 | 0 (0 to 50) | 50 | 0 (0 to 50) |
| **Governance** | 14 | 76 | 1 (-10 to 10) | 76 | 8 (-8 to 20) | 76 | 6 (-14 to 10) | 82 | -5 (-11 to 5) | 71 | -7 (-11 to 7) |

green >15% increase
light green 5-15% increase
orange -5 < & < 5
light red 5 to -15% decrease
dark red >15% decrease

**Fig 3. Heat map of median standards-based service readiness scores at baseline, percentage point change in median, and interquartile range of percent change by HSBB and by HSBB sub-module for 65 NEST360 implementing neonatal units.** Abbreviations: MPDSR – Maternal and perinatal death surveillance and response.

prevention readiness [Median 67%, IQR 0–67%] (Fig 4). In Tanzania, increased readiness for this intervention was also driven by improvements in blood glucose screening [Median 100%, IQR 67–100%], cup feeding [Median 33%, IQR 0–33%], IV fluids [Median 33%, IQR 0–67%], and guideline and training readiness [Median 67%, IQR 0–100%]. Similarly, in Nigeria, improvements were also driven by blood glucose screening [Median 67%, IQR 0–100%], NG tube feeding [Median 33%, IQR 0–100%], and device power source readiness [Median 34%, IQR 0–34%]. In all hospitals, change in detection and management of jaundice with phototherapy, and management of exchange transfusion, which require some similar items, was driven by improvements in bilirubin measurement [Median 33%, IQR 0–67%] and infection prevention readiness [Median 67%, IQR 0–67%], as well as guideline and training readiness for phototherapy use [Median 100%, IQR 0–100%]. In hospitals in Nigeria, which showed the largest change in phototherapy readiness, change was also driven by improvements in laboratory capacity to assess underlying causes [Median 33%, IQR 0–0%] and device power source readiness [Median 34%, IQR 0–67%]. Overall, safe administration of oxygen improvement was driven by increased oxygen assessment [Median 34%, IQR 0–67%] and infection prevention readiness [Median 67%, IQR 0–67%]. In hospitals in Nigeria, this improvement was also driven by increased readiness of items for oxygen provision [Median 100%, IQR 33–67%], oxygen sources including devices [Median 34%, IQR 34–67%], and oxygen power sources [Median 34%, IQR 0–34%].

| | n items | Total | | Malawi | | Kenya | | Tanzania | | Nigeria | |
|---|---|---|---|---|---|---|---|---|---|---|---|
| | | Baseline | % change (IQR) | Baseline | % change (IQR) | Baseline | % change (IQR) | Baseline | % change (IQR) | Baseline | % change (IQR) |
| **Overall** | 309 | 41 | 14 (4 to 18) | 41 | 8 (-2 to 12) | 50 | 7 (2 to 15) | 41 | 24 (16 to 30) | 33 | 28 (17 to 30) |
| **Thermal care, including Kangaroo Mother Care (KMC)** | 59 | 46 | 4 (-8 to 17) | 46 | 0 (-8 to 6) | 50 | 0 (-4 to 8) | 33 | 21 (8 to 33) | 29 | 25 (21 to 25) |
| Temperature monitoring for baby | 5 | 67 | 0 (-33 to 0) | 100 | -33 (-33 to 0) | 100 | -33 (-33 to 0) | 67 | 0 (0 to 0) | 67 | 33 (0 to 100) |
| Items for thermal support | 3 | 33 | 0 (0 to 34) | 0 | 33 (0 to 33) | 33 | 34 (0 to 34) | 0 | 67 (0 ot 67) | 33 | 67 (0 to 67) |
| Kangaroo Mother Care | 7 | 33 | 0 (0 to 0) | 33 | 0 (0 to 0) | 33 | 0 (0 to 34) | 33 | 0 (-34 to 0) | 0 | 0 (0 to 33) |
| Devices for thermal support | 10 | 67 | 0 (0 to 33) | 33 | -33 (-33 to 0) | 67 | 0 (0 to 0) | 33 | 34 (0 to 67) | 67 | 0 (0 to 34) |
| Device power sources | 14 | 67 | -34 (-34 to 0) | 67 | -34 (-34 to 0) | 33 | 0 (-34 to 0) | 67 | -34 (-34 to 0) | 33 | 34 (0 to 34) |
| Infection prevention | 11 | 0 | 67 (0 to 67) | 0 | 67 (0 to 67) | 67 | 0 (0 to 67) | 0 | 67 (0 to 67) | 0 | 67 (0 to 67) |
| Infrastructure for thermal support | 4 | 33 | -33 (-33 to 0) | 33 | -33 (-100 to 0) | 33 | -33 (-33 to -33) | 0 | 0 (0 to 0) | 0 | 0 (0 to 0) |
| Guidelines, initiation of care, and training | 5 | 33 | 67 (0 to 67) | 33 | 67 (0 to 67) | 100 | 0 (0 to 67) | 0 | 100 (0 to 100) | 33 | 67 (-33 to 67) |
| **Assisted feeding and intravenous fluids** | 107 | 33 | 11 (7 to 22) | 30 | 7 (4 to 15) | 44 | 15 (4 to 19) | 44 | 34 (30 to 44) | 26 | 33 (11 to 37) |
| Blood glucose screening | 13 | 0 | 33 (0 to 67) | 0 | 0 (-16 to 0) | 33 | 67 (0 to 67) | 0 | 100 (67 to 100) | 0 | 67 (0 to 100) |
| Breast feeding & milk banking | 8 | 0 | 0 (0 to 0) | 0 | 0 (0 to 0) | 0 | 0 (0 to 0) | 0 | 0 (0 to 0) | 0 | 0 (0 to 33) |
| Cup feeding | 6 | 67 | 0 (0 to 33) | 67 | 0 (-50 to 16) | 67 | 0 (-33 to 0) | 67 | 33 (0 to 33) | 67 | 33 (0 to 100) |
| Nasogastric tube feeding | 8 | 67 | 0 (0 to 0) | 67 | 0 (0 to 0) | 67 | 0 (-33 to 0) | 67 | 0 (0 to 33) | 67 | 33 (0 to 100) |
| Intravenous fluids | 9 | 33 | 0 (0 to 33) | 33 | 0 (0 to 33) | 33 | 67 (0 to 67) | 67 | 33 (0 to 67) | 100 | -67 (-67 to 0) |
| Intravenous fluids equipment and consumables | 10 | 0 | 67 (0 to 67) | 0 | 67 (0 to 67) | 33 | 34 (0 to 67) | 0 | 100 (33 to 100) | 0 | 67 (34 to 67) |
| Device power sources | 14 | 67 | -34 (-34 to 0) | 67 | -17 (-34 to 0) | 33 | 0 (-34 to 0) | 67 | -34 (-34 to 0) | 33 | 34 (0 to 34) |
| Infection prevention | 15 | 0 | 67 (0 to 67) | 0 | 67 (0 to 67) | 67 | 0 (-33 to 67) | 0 | 100 (-33 to 100) | 0 | 67 (0 to 67) |
| Guidelines, initiation of care, and training | 24 | 0 | 0 (0 to 33) | 0 | 0 (0 to 0) | 0 | 67 (0 to 67) | 33 | 67 (0 to 100) | 0 | 33 (0 to 34) |
| **Safe administration of oxygen** | 64 | 43 | 14 (5 to 24) | 43 | 14 (5 to 21) | 48 | 9 (0 to 19) | 33 | 24 (14 to 24) | 29 | 33 (24 to 38) |
| Oxygen assessment | 5 | 33 | 34 (0 to 67) | 67 | 0 (0 to 67) | 67 | 0 (0 to 67) | 0 | 0 (0 to 67) | 0 | 67 (0 to 67) |
| Vital sign monitoring | 7 | 67 | 0 (0 to 0) | 67 | 0 (0 to 0) | 67 | 0 (-33 to 0) | 67 | 0 (0 to 0) | 67 | 0 (0 to 67) |
| Items for oxygen provision | 15 | 33 | 34 (0 to 34) | 33 | 0 (0 to 33) | 67 | 0 (0 to 34) | 0 | 100 (33 to 100) | 0 | 100 (33 to 67) |
| Oxygen sources, including devices | 5 | 33 | 34 (0 to 34) | 33 | 0 (0 to 34) | 67 | 0 (0 to 33) | 33 | 0 (0 to 34) | 33 | 34 (34 to 67) |
| Device power sources | 14 | 67 | -34 (-34 to 0) | 67 | -17 (-34 to 0) | 33 | 0 (-34 to 0) | 67 | -34 (-67 to 0) | 33 | 34 (0 to 34) |
| Infection prevention | 12 | 0 | 67 (0 to 67) | 0 | 67 (0 to 67) | 67 | 0 (0 to 67) | 0 | 67 (-33 to 67) | 0 | 67 (0 to 67) |
| Guidelines, initiation of care, and training | 6 | 0 | 67 (0 to 67) | 0 | 33 (0 to 67) | 0 | 67 (-33 to 67) | 33 | 67 (0 to 100) | 100 | 0 (-67 to 67) |
| **Neonatal sepsis with antibiotics** | 42 | 58 | 0 (-17 to 8) | 58 | -8 (-17 to 8) | 75 | -8 (-33 to 0) | 75 | 0 (-17 to 8) | 42 | 8 (-8 to 33) |
| Readiness for culture | 17 | 0 | 0 (0 to 0) | 0 | 0 (0 to 0) | 0 | 33 (0 to 0) | 100 | -33 (-33 to 67) | 33 | 0 (0 to 34) |
| Temperature monitoring for baby | 5 | 67 | 0 (-33 to 0) | 100 | -33 (-33 to 0) | 100 | -33 (-33 to 0) | 67 | 0 (0 to 67) | 67 | 33 (0 to 100) |
| Antibiotics | 13 | 100 | 0 (0 to 0) | 100 | 0 (0 to 0) | 100 | 0 (0 to 0) | 100 | 0 (0 to 67) | 33 | 0 (0 to 0) |
| Guidelines, initiation of care, and training | 7 | 33 | 0 (-33 to 0) | 33 | 0 (-33 to 33) | 100 | -33 (-33 to -33) | 100 | -33 (-33 to -33) | 33 | 0 (-33 to 33) |
| **Neonatal jaundice with phototherapy** | 69 | 37 | 19 (7 to 22) | 41 | 13 (4 to 19) | 33 | 19 (7 to 30) | 37 | 22 (7 to 30) | 33 | 30 (19 to 33) |
| Bilirubin measurement | 14 | 0 | 33 (0 to 67) | 0 | 0 (0 to 67) | 0 | 33 (0 to 100) | 0 | 67 (67 to 100) | 0 | 67 (0 to 100) |
| Lab can assess underlying causes | 7 | 0 | 0 (0 to 0) | 0 | 0 (0 to 0) | 0 | 0 (0 to 0) | 0 | 0 (0 to 0) | 0 | 33 (0 to 0) |
| Other monitoring for baby | 8 | 67 | 0 (-33 to 0) | 100 | -33 (-33 to 0) | 100 | -33 (-33 to 0) | 67 | 0 (0 to 0) | 67 | 0 (0 to 100) |
| Equipment for phototherapy provision | 4 | 67 | 33 (0 to 33) | 67 | 33 (0 to 33) | 67 | 0 (0 to 33) | 100 | 0 (0 to 0) | 100 | 0 (0 to 33) |
| Consumables for phototherapy provision | 4 | 0 | 0 (0 to 0) | 0 | 0 (0 to 0) | 0 | 0 (0 to 0) | 0 | 0 (0 to 0) | 0 | 0 (0 to 0) |
| Device power sources | 14 | 67 | -34 (-34 to 0) | 67 | -34 (-34 to 0) | 33 | 0 (-34 to 0) | 67 | -34 (-34 to 0) | 33 | 34 (0 to 34) |
| Therapeutic irradiance | 1 | 100 | 0 (0 to 0) | 100 | 0 (0 to 0) | 100 | 0 (0 to 67) | 0 | 0 (-100 to 100) | 100 | 0 (0 to 100) |
| Infection prevention | 11 | 0 | 67 (0 to 67) | 0 | 67 (0 to 67) | 67 | 0 (0 to 67) | 0 | 67 (0 to 67) | 0 | 67 (0 to 67) |
| Guidelines, initiation of care, and training | 6 | 0 | 100 (0 to 100) | 33 | 67 (0 to 100) | 0 | 67 (-33 to 67) | 0 | 100 (0 to 100) | 33 | 34 (0 to 67) |
| **Neonatal encephalopathy** | 29 | 25 | 31 (0 to 39) | 21 | 23 (-15 to 38) | 33 | 23 (3 to 36) | 17 | 61 (19 to 69) | 17 | 16 (0 to 39) |
| Diagnostics | 18 | 0 | 33 (0 to 34) | 0 | 0 (-16 to 0) | 33 | 34 (0 to 34) | 0 | 100 (34 to 100) | 0 | 33 (0 to 67) |
| Seizure management | 6 | 33 | 0 (-34 to 0) | 33 | 0 (-34 to 0) | 33 | 0 (0 to 0) | 33 | 0 (0 to 0) | 33 | 0 (-34 to 0) |
| Guidelines, initiation of care, and training | 5 | 33 | 67 (0 to 100) | 33 | 67 (0 to 100) | 0 | 100 (0 to 100) | 33 | 67 (0 to 100) | 0 | 0 (0 to 67) |
| **Referral of congenital abnormalities** | 23 | 56 | 11 (0 to 22) | 56 | 11 (-17 to 17) | 67 | 22 (0 to 22) | 56 | 44 (0 to 44) | 56 | 11 (0 to 33) |
| Referral communication systems | 10 | 67 | 33 (0 to 33) | 67 | 0 (0 to 33) | 100 | 0 (0 to 33) | 100 | 0 (0 to 33) | 67 | 33 (33 to 33) |
| Referral transport systems | 11 | 100 | 0 (0 to 0) | 100 | 0 (0 to 0) | 67 | 0 (0 to 0) | 100 | 0 (-33 to 0) | 100 | -33 (0 to 0) |
| Guidelines, initiation of care, and training | 2 | 0 | 67 (0 to 67) | 0 | 0 (-33 to 0) | 100 | 0 (0 to 67) | 0 | 100 (67 to 100) | 0 | 67 (0 to 100) |
| **CPAP management of preterm resp. distress** | 80 | 46 | 12 (4 to 25) | 50 | 8 (2 to 12) | 46 | 12 (4 to 25) | 33 | 25 (0 to 29) | 21 | 41 (29 to 46) |
| Oxygen assessment | 5 | 33 | 34 (0 to 67) | 67 | 0 (0 to 67) | 67 | 0 (0 to 67) | 0 | 0 (0 to 67) | 0 | 67 (0 to 67) |
| Vital sign monitoring | 8 | 67 | 0 (0 to 0) | 67 | 0 (0 to 0) | 67 | 0 (-33 to 0) | 67 | 0 (0 to 0) | 67 | 0 (0 to 67) |
| Items for CPAP | 26 | 33 | 34 (0 to 67) | 50 | 17 (0 to 34) | 0 | 67 (0 to 67) | 0 | 33 (0 to 33) | 0 | 100 (33 to 100) |
| Equipment for CPAP | 4 | 0 | 67 (0 to 33) | 0 | -16 (0 to 0) | 67 | 0 (0 to 67) | 67 | 33 (-33 to 33) | 0 | 100 (0 to 100) |
| Oxygen sources, including devices | 5 | 33 | 34 (0 to 34) | 33 | 0 (0 to 34) | 67 | 0 (0 to 34) | 33 | 0 (0 to 34) | 33 | 34 (34 to 67) |
| Device power sources | 14 | 67 | -34 (-34 to 0) | 67 | -34 (-34 to 0) | 33 | 0 (-34 to 0) | 67 | -34 (-34 to 0) | 33 | 34 (0 to 34) |
| Infection prevention | 12 | 0 | 67 (0 to 67) | 0 | 67 (0 to 67) | 67 | 0 (0 to 67) | 0 | 67 (-33 to 67) | 0 | 67 (0 to 67) |
| Guidelines, initiation of care, and training | 6 | 100 | 0 (0 to 67) | 100 | 0 (0 to 0) | 0 | 67 (0 to 67) | 0 | 100 (0 to 100) | 33 | 34 (0 to 67) |
| **Exchange transfusion for a newborn** | 63 | 24 | 14 (5 to 24) | 19 | 19 (5 to 24) | 33 | 5 (5 to 24) | 33 | 19 (19 to 33) | 29 | 23 (14 to 29) |
| Bilirubin measurement | 14 | 0 | 33 (0 to 67) | 0 | 0 (0 to 67) | 0 | 33 (0 to 100) | 0 | 67 (67 to 100) | 0 | 67 (0 to 100) |
| Lab can assess underlying causes | 7 | 0 | 0 (0 to 0) | 0 | 0 (0 to 0) | 0 | 0 (0 to 0) | 0 | 0 (0 to 0) | 0 | 33 (0 to 0) |
| Other monitoring for baby | 8 | 67 | 0 (-33 to 0) | 100 | -33 (-33 to 0) | 100 | -33 (-33 to 0) | 67 | 0 (0 to 0) | 67 | 0 (0 to 100) |
| Equipment for exchange transfusion | 11 | 0 | 0 (0 to 0) | 0 | 0 (0 to 0) | 0 | 0 (0 to 0) | 0 | 0 (0 to 0) | 0 | 0 (0 to 0) |
| Blood bank support for transfusion | 8 | 33 | 34 (0 to 34) | 33 | 34 (0 to 34) | 33 | 0 (0 to 33) | 67 | -34 (-34 to 0) | 67 | 0 (-33 to 0) |
| Infection prevention | 13 | 0 | 67 (0 to 67) | 0 | 67 (0 to 67) | 67 | 0 (-33 to 67) | 0 | 100 (-33 to 100) | 0 | 67 (0 to 67) |
| Guidelines, initiation of care, and training | 2 | 0 | 33 (0 to 100) | 0 | 0 (0 to 33) | 0 | 100 (0 to 100) | 0 | 100 (33 to 100) | 33 | 67 (0 to 67) |
| **Follow-up of at-risk newborns** | 3 | 33 | 0 (-33 to 33) | 33 | 0 (-33 to 33) | 100 | 0 (0 to 67) | 0 | 67 (0 to 100) | 33 | 0 (-33 to 67) |
| Guidelines and discharge plan | 3 | 33 | 0 (-33 to 33) | 33 | 0 (-33 to 33) | 100 | 0 (0 to 67) | 0 | 67 (0 to 100) | 33 | 0 (-33 to 67) |

**Legend:**
- green >15% increase
- light green 5-15% increase
- orange -5 < & < 5
- light red 5 to -15% decrease
- dark red >15% decrease

**Fig 4. Heat map of median level-2+ service readiness scores at baseline, percentage point change in median, and interquartile range of percent change, by intervention and by intervention sub-module for 65 NEST360 implementing neonatal units.** Abbreviations: CPAP – Continuous Positive Airway Pressure.

# Discussion

Tracking changes and understanding the drivers of change in service readiness are vital. However, few studies have assessed service readiness at multiple time points, and we found no published studies of changes over time for level-2+neonatal care in low- and middle-income countries (LMICs). In this analysis of 65 neonatal units in Kenya, Malawi, Nigeria, and Tanzania, service readiness for level-2+ [Median 14%, IQR 4–18%] and standards-based [Median 9%, IQR 3–11%] SSNC increased from baseline, prior to NEST360 implementation, to mid-2023. Hospitals in Tanzania showed greater improvements in standards-based scores, and hospitals in Tanzania and Nigeria showed greater improvements in level-2+ scores compared with the hospitals included in Kenya and Malawi.

For all hospitals, percentage change was somewhat higher for HSBBs that were a focus of NEST360 implementation, particularly human resources [Median 13%, IQR 5–18%], information systems [Median 11%, IQR 2–15%], and medical devices and supplies [Median 9%, IQR 5–13%]. Improvements in medical devices and supplies were particularly driven by increased availability of equipment and consumables for level-2+clinical interventions on the neonatal unit, and increased readiness for the biomedical workshop, which includes device maintenance and systems improvements for the hospital and was supported through the NEST360 Alliance [19]. Information systems was also a priority for NEST360, including data management training and patient-level data collection system set-up, which were provided to support the development of high-quality routine data systems for newborn care and data for decision-making [20]. Human resources, particularly education and training for clinicians, biomedical technicians, and data clerks, was also supported by the NEST360 programme [21]. Jointly with government partners, NEST360 teams led generic instructor courses (GICs) to expand availability of high-quality trainers for clinical and biomedical education in countries implementing with NEST360 [22]. Although there were improvements in the human resources scores, the standards-based scores focused on people, education, and enabling environment, and did not include staff-to-patient ratios, which currently have no standards, and remain a challenge across hospitals in the four countries. In addition, the people sub-module, which focuses on staffing allocation and staff numbers, did not improve, which suggests the need for additional government investment to strengthen the neonatal health workforce by recruiting, training, and retaining neonatal unit staff and supporting cadres, including clinical specialists. Infrastructure in hospitals in Nigeria and Tanzania showed great improvements, which may be related to significant government co-investment and support for neonatal unit infrastructure improvements. For example, at two regional referral hospitals in Tanzania, the government invested in redesign of the entire newborn ward, improving electrical power readiness and ward space and design, particularly for workflow in clinical care spaces and improvements in non-care spaces, such as staff break areas [23]. Hospitals in Malawi showed greater improvements in governance and leadership readiness, which may be related to a quality improvement programme aimed at enhancing hospital management practices in support of high-quality newborn care [24].

The percent change in level-2+ intervention scores was higher but more variable than for standards-based scores, and may be more discriminatory. Changes were more pronounced for the detection and management of neonatal encephalopathy [Median 31%, IQR 0–40%], detection and management of jaundice with phototherapy [Median 19%, IQR 7–22%], safe administration of oxygen [Median 14%, IQR 5–24%], and management of exchange transfusion [Median 14%, IQR 5–24%] compared to other interventions. Improvements in many of these interventions were driven by increased infection prevention readiness and increased readiness to screen and detect clinical symptoms, and may be linked to wider health systems improvements in infection prevention and availability of technologies for clinical screening and detection. These changes may also be associated with the NEST360 programme's emphasis on availability and maintenance of equipment and consumables, as well as expanded education for level-2+ clinical interventions [19,22].

Standards-based scores focus on health system readiness in support of the newborn care unit, and have a wider remit assessing readiness across the hospital. These changes in scores can be useful for tracking improvements by HSBB, particularly when a service or department provides support across the hospital. For example, improvements in biomedical

workshop readiness would not be directly tracked as part of level-2+ clinical intervention scores, however, improvements in biomedical workshop readiness, including availability of spare parts and tools, preventive and corrective maintenance systems, and trained biomedical technicians, are vital to ensure that devices in the newborn ward are well maintained for use in clinical care. The level-2+ clinical intervention scores are also useful and provide information about the essential items needed for level-2 and transition clinical interventions. These scores include fewer items and may be easily integrated into routine systems for tracking over time. While there is some overlap in standards-based and level-2+ intervention scores, level-2+ scores provide different information about the gaps. For example, readiness for bilirubin measurement may only include a few items, and therefore might be challenging to assess in larger HSBBs using standards-based scoring. However, the ability to measure bilirubin is vital for assessing the need for phototherapy or exchange transfusions. The level-2+ scores provide clear scores to assess the gaps in readiness to assess, detect, manage, and treat level-2+ clinical interventions. However, if a hospital performs poorly on level-2+ interventions, it may be helpful to look at the wider health system using standards-based scores, as wider hospital readiness is likely to inform service readiness on the newborn unit.

Published studies on service readiness for child health or for essential newborn care signal functions report varied findings from minor or significant change to minimal or negative change [12–14]. A small programme in India assessed change in readiness for child health services during a one-year time period, and noted that infrastructure readiness components did not change, which is consistent with our overall results [13]. Other studies highlight variations across countries. For example, one study, which assessed readiness to provide maternal and newborn care signal functions in facilities in Nigeria, Ethiopia, and India, noted decreases, increases, and no change respectively in facility readiness to provide clean cord care for newborns (a signal function) between 2012 and 2015 [12]. A national study in Nepal assessed readiness to provide basic emergency obstetric signal functions and one for newborn (neonatal resuscitation), reporting no significant change in service readiness over a 6 year period [14].

Our evaluation had several strengths, notably the use of a standard tool and comparable analyses. Data collection at baseline and 2023 follow-up visits were consistent, as the same HFA tool was used at both time points with minor adaptations at 2023 visits, and most data collection teams and assessors remained the same. We collected data from a large number of neonatal units across four African countries (Kenya, Malawi, Nigeria, and Tanzania) at two separate time points several years apart. We applied two scoring approaches to aggregate HFA readiness data considering HSBBs or clinical intervention, each with sub-modules, enabling more specific identification of gaps and opportunities for improvement and collaborative learning, as well as tracking progress over time.

Although hospitals may be more ready to provide high-quality care, the observed increase in service readiness may not result in improvements in direct clinical care or patient-level outcomes. The interventions included in the scoring were equally weighted in these scores, yet some items may be more important than others for improving patient-level outcomes. It may be useful to develop weighted scores; however, this would require analyses linked to clinical coverage and patient outcomes. We note that although the assessments were conducted at the national scale in Malawi, the findings do not include all hospitals in Kenya, Nigeria, and Tanzania, so they are not expected to be representative of these countries. It is also not possible to attribute all changes in the HFAs to the NEST360 implementation programme alone, as there were other changes at the hospital-level during this time period, and it was an explicit aim of the NEST360 alliance to catalyse more change. There were also negative impacts in this time, notably the COVID pandemic and climate disruptions, particularly in Malawi [25].

These results demonstrate the value of tracking service readiness over time, especially for sub-national and national governments, to inform progress towards targets. It may be desirable to incorporate some service readiness indicators, such as staff-to-baby ratios, into routine systems for more frequent tracking while also collecting service readiness data on the wider neonatal unit and hospital system to address gaps for more stagnant indicators. Our results also demonstrate the need for further investment in health systems for SSNC. For example, infrastructure for newborn units may need

further investments to achieve measurable change, as demonstrated by the example of a referral hospital in Tanzania [26]. For human resources, although changes at these hospitals were demonstrated particularly with regard to education and the enabling environment, additional investment and resources are needed to strengthen the health workforce and manage health worker's workload in the long term.

Future work could further explore whether it is possible to conduct repeated service readiness assessments for SSNC using a further reduced dataset, noting this tool was designed to be used in less than one day [7]. In addition, it would be important to evaluate whether changes in service readiness are associated with changes in patient-level outcomes, such as mortality. Weighted scores could then be developed to account for relative importance of specific items to predict clinical care and patient outcomes. Analyses of service indicators, such as staff-to-baby ratios, would be helpful to inform development of standards of care. Though it was not possible to include in these analyses, later work could explore in-depth drivers of changes in service readiness at a facility and national level to identify drivers of change which may be applied to other programmes.

## Conclusion

Our evaluation demonstrates that it is possible to improve readiness for high-quality small and sick newborn care, even over relatively short time periods, and during a pandemic. Baseline standards-based and level-2+ scores were low across all countries. There were major improvements, especially in readiness for some HSBBs and level-2+ clinical interventions. Despite progress, there are still large gaps between hospital service readiness and the standards of care. Though some aspects of service readiness for SSNC may be more challenging and require additional investment, notably infrastructure, improvements are possible [26]. To achieve global newborn care targets and reduce the 2.3 million newborn deaths per year, more ambition is needed so that hospitals are ready to provide higher-quality care.

## Supporting information

**S1 File. Baseline and 2023 Health Facility Assessment (HFA) data collection dates by hospital and country.** (PDF)

**S2 File. Standards-based and level-2+ clinical interventions and sub-modules included in Health Facility Assessment (HFA) scoring.** (DOCX)

**S3 File. Local ethical approval numbers for the complex evaluation of the implementation of a small and sick newborn care package with Newborn Essential Solutions and Technologies (NEST360).** (DOCX)

**S4 File. Hospital level and admissions for 65 neonatal units implementing with Newborn Essential Solutions and Technologies (NEST360) in Kenya, Malawi, Nigeria, and Tanzania.** (DOCX)

**S5 File. Spider plot of median standards-based scores by health system building blocks and level-2+ scores by clinical intervention for 65 neonatal units implementing with Newborn Essential Solutions and Technologies (NEST360) at baseline and 2023 follow-up with separate country-specific panels.** Legend: Abbreviations: IV – Intravenous; CPAP – Continuous Positive Airway Pressure. (TIF)

**S1 Checklist. Inclusivity in global research checklist.** (DOCX)

## Acknowledgments

First, and most importantly, we thank the newborns and their mothers whose data are at the heart of NEST360. We also thank those involved as part of the NEST360 HFA development and all the data teams, health workers, and others involved in collecting and using the data. We thank the relevant administrative staff for their assistance.

## Author contributions

**Conceptualization:** Rebecca E Penzias, Joy E. Lawn, Eric O. Ohuma.

**Data curation:** Rebecca E Penzias, Morris Ondieki Ogero, James H. Cross, Eric O. Ohuma.

**Formal analysis:** Rebecca E Penzias, Christine Bohne, David Gathara, Simon Cousens, Joy E. Lawn, Eric O. Ohuma.

**Funding acquisition:** Honorati Masanja, Nahya Salim, Veronica Chinyere Ezeaka, William M. Macharia, Msandeni Chiume, Queen Dube, Elizabeth M. Molyneux, Maria Oden, Rebecca Richards-Kortum, Joy E. Lawn.

**Investigation:** Rebecca E Penzias, Morris Ondieki Ogero, Robert Tillya, Irabi Kassim, Olabisi Dosunmu, Opeyemi Odedere, Hannah Mwaniki, Vincent O. Ochieng, Dolphine Mochache, Samuel K. Ngwala, Evelyn Zimba, Grace T. Soko, Christine Bohne, James H. Cross, Josephine Shabani, Catherine Paul, Donat Shamba, Nahya Salim, Charles Osuagwu, Afeez Idowu, Ifeanyichukwu Anthony Ogueji, Olukemi Tongo, Olabanjo Okunlola Ogunsola, Veronica Chinyere Ezeaka, Ekran Rashid, George Okello, John Wainaina, Msandeni Chiume, Alfred Chalira, Edith Gicheha, Millicent Alooh.

**Methodology:** Rebecca E Penzias, Simon Cousens, Joy E. Lawn, Eric O. Ohuma.

**Project administration:** Rebecca E Penzias, Morris Ondieki Ogero, Robert Tillya, Irabi Kassim, Olabisi Dosunmu, Opeyemi Odedere, Hannah Mwaniki, Vincent O. Ochieng, Dolphine Mochache, Samuel K. Ngwala, Evelyn Zimba, Eric O. Ohuma.

**Software:** Rebecca E Penzias, Morris Ondieki Ogero.

**Supervision:** Opeyemi Odedere, Evelyn Zimba, Catherine Paul, Honorati Masanja, Nahya Salim, Veronica Chinyere Ezeaka, George Okello, William M. Macharia, Msandeni Chiume, Queen Dube, Simon Cousens, Maria Oden, Rebecca Richards-Kortum, Joy E. Lawn, Eric O. Ohuma.

**Visualization:** Rebecca E Penzias, Christine Bohne, David Gathara, Simon Cousens, Joy E. Lawn, Eric O. Ohuma.

**Writing – original draft:** Rebecca E Penzias.

**Writing – review & editing:** Rebecca E Penzias, Morris Ondieki Ogero, Robert Tillya, Irabi Kassim, Olabisi Dosunmu, Opeyemi Odedere, Hannah Mwaniki, Vincent O. Ochieng, Dolphine Mochache, Samuel K. Ngwala, Evelyn Zimba, Grace T. Soko, Christine Bohne, David Gathara, James H. Cross, Josephine Shabani, Catherine Paul, Donat Shamba, Honorati Masanja, Nahya Salim, Charles Osuagwu, Afeez Idowu, Ifeanyichukwu Anthony Ogueji, Olukemi Tongo, Olabanjo Okunlola Ogunsola, Veronica Chinyere Ezeaka, Ekran Rashid, George Okello, John Wainaina, William M. Macharia, Msandeni Chiume, Alfred Chalira, Queen Dube, Edith Gicheha, Elizabeth M. Molyneux, Millicent Alooh, Simon Cousens, Maria Oden, Rebecca Richards-Kortum, Joy E. Lawn, Eric O. Ohuma.

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
