## [Decision Letter · Decision Letter 0]

PGPH-D-24-01799

Small and sick newborn care: Changes in service readiness scoring between baseline and 2023 for 63 hospitals implementing with NEST360 in Kenya, Malawi, Nigeria, and Tanzania

Dear Dr. Penzias,

Thank you for submitting your manuscript to PLOS Global Public Health. After careful consideration, we feel that it has merit but does not fully meet PLOS Global Public Health’s publication criteria as it currently stands. Therefore, we invite you to submit a revised version of the manuscript that addresses the points raised during the review process.

We look forward to receiving your revised manuscript.

Kind regards,

Oghenebrume Wariri, MD, PhD

Academic Editor

Journal Requirements:

**Please only choose the relevant sentences from below**

1. Please clarify all sources of funding (financial or material support) for your study. List the grants (with grant number) or organizations (with url) that supported your study, including funding received from your institution. 

2. State the initials, alongside each funding source, of each author to receive each grant.

3. State what role the funders took in the study. If the funders had no role in your study, please state: “The funders had no role in study design, data collection and analysis, decision to publish, or preparation of the manuscript.”

4. If any authors received a salary from any of your funders, please state which authors and which funders

3. Please send a completed 'Competing Interests' statement, including any COIs declared by your co-authors. If you have no competing interests to declare, please state "The authors have declared that no competing interests exist". Otherwise please declare all competing interests beginning with the statement "I have read the journal's policy and the authors of this manuscript have the following competing interests:

4. In the online submission form, you indicated that "All partners in the NEST360 alliance collaborated to create and sign data sharing and transfer agreements. The dataset from this study will be accessible upon request, pending approval from the NEST360 learning network and collaborating parties.". 

3. Uploaded as supplementary information.

5. Please provide an Author Summary. This should appear in your manuscript between the Abstract (if applicable) and the Introduction, and should be 150–200 words long. The aim should be to make your findings accessible to a wide audience that includes both scientists and non-scientists. Sample summaries can be found on our website under Submission Guidelines: 

https://journals.plos.org/globalpublichealth/s/submission-guidelines#loc-parts-of-a-submission

6. We have noticed that you have uploaded Supporting Information files, but you have not included a list of legends. Please add a full list of legends for your Supporting Information files after the references list.

Additional Editor Comments (if provided):

Thank you for submiting this manuscript reporting this important piece of work. Kindly address all the comments raised by the two reviewers.

Reviewers' comments:

Reviewer's Responses to Questions

**Comments to the Author**

1. Does this manuscript meet PLOS Global Public Health’s publication criteria?

Reviewer #1: Yes

Reviewer #2: Yes

2. Has the statistical analysis been performed appropriately and rigorously?

Reviewer #1: Yes

Reviewer #2: Yes

3. Have the authors made all data underlying the findings in their manuscript fully available (please refer to the Data Availability Statement at the start of the manuscript PDF file)?

Reviewer #1: Yes

Reviewer #2: Yes

4. Is the manuscript presented in an intelligible fashion and written in standard English?

Reviewer #1: Yes

Reviewer #2: Yes

Reviewer #1: Thank you for asking me to review this well-written paper on changes in service readiness for small and sick newborns (SSNC). I have a few suggestions for the authors:

Major revision

1. It is difficult to fully understand the paper without reading your other 2 papers - reference 7 & 8. I think a neutral reader should be able to pick up this paper and read to understand without having to visit the other papers. A few suggestion to improve on this -

a. line 90 and 91. Maybe a few words on the scoring approaches for some background. I assume these are based on the HSBB and level 2+ clinical intervention. With the way the paper is currently written, there is no introduction to this before they come up in the aims. This makes your aims difficult to understand without reading your 2 papers or going into reading your methodology

b. The HFA can be better described - how many domains/ questions does it assess? I found this in another paper. The authors speak about modules and sub-modules later in the methods. A gentle introduction on how the HFA was structured will allow the reader understand this without having to visit the other linked papers.

c. For such a detailed HFA completed in under 24 hours - how and to who was this administered and what methods were used? The questions span what nurses or lab manager would know and seems to cut across multiple hospital departments linked to SSNC.

Minor revisions -

1.Results speak about 65 neonatal units but title and abstract talk about 63. why the discrepancy?

2. Please in the methods, highlight the continuous quality improvement linked to NEST-360, so the reader appreciates the intervention was not one-off and was sustained over a period of time.

3. Was the decision to analyse Malawi as low volume and high volume determined apriori or this was after a look at the data? The authors should be transparent about this in the methods

4. Although human resources had the highest improvements between periods, this was driven by education & training and not an increase in people. The authors should reflect on this in the discussion including any possible implications. Currently there is some reflection on what was not measured (nurse to patient ratios), but you have a whole subdomain on people in the HSBB which did not change much between the baseline and after NEST.

Reviewer #2: Thank you for submitting this well-written manuscript on an important and timely topic related to the global neonatal survival target and Every Newborn Action Plan.

The study aim(s) are clearly stated (lines 113-118) as (i) evaluating changes in standards-based and level-2+ service readiness between baseline (2019-2021) and 2023 follow-up scores for each neonatal unit according to Health System Building Block (HSBB) or clinical intervention, and (ii) evaluating changes in standards-based and level-2+ service readiness for each neonatal unit according to HSBB or clinical intervention disaggregated by sub

module scores.

It seems as though an additional aim of this study was to explore factors that contributed to (or perhaps were barriers to) improvements in service readiness as measured by the HFA score. On line 360, you state: "Tracking changes and understanding the drivers of change in service readiness are vital." The "Discussion" section of your manuscript also explores various drivers of change in service readiness.

If possible, I suggest that you clearly state an additional aim, "Exploring the drivers of change in service readiness," as it appears that this was one of the aims of your research. Adding a subsection on the drivers of change in service readiness would allow you to clearly describe (perhaps country by country) potential drivers of change in the HFA service readiness score.

Adding this additional explicit aim, which seems to be implicitly part if this research, and more discussion on the drivers of change, could strengthen the applicability and usefulness of this important research.

**Do you want your identity to be public for this peer review?** For information about this choice, including consent withdrawal, please see our Privacy Policy

Reviewer #1: No

Reviewer #2: **Yes: ** Laura Miniea Hoemeke

---

## [Decision Letter · Decision Letter 1]

Small and sick newborn care: Changes in service readiness scoring between baseline and 2023 for 65 neonatal units implementing with NEST360 in Kenya, Malawi, Nigeria, and Tanzania

PGPH-D-24-01799R1

Dear Ms Penzias,

We are pleased to inform you that your manuscript 'Small and sick newborn care: Changes in service readiness scoring between baseline and 2023 for 65 neonatal units implementing with NEST360 in Kenya, Malawi, Nigeria, and Tanzania' has been provisionally accepted for publication in PLOS Global Public Health.

Best regards,

Julia Robinson

Executive Editor

Reviewer Comments (if any, and for reference):

Reviewer's Responses to Questions

**Comments to the Author**

Reviewer #1: All comments have been addressed

publication criteria?

Reviewer #1: Yes

3. Has the statistical analysis been performed appropriately and rigorously?

Reviewer #1: Yes

4. Have the authors made all data underlying the findings in their manuscript fully available (please refer to the Data Availability Statement at the start of the manuscript PDF file)?

Reviewer #1: No

5. Is the manuscript presented in an intelligible fashion and written in standard English?

Reviewer #1: Yes

Reviewer #1: No further comments. Authors have fully answered all my earlier queries.

**Do you want your identity to be public for this peer review?** For information about this choice, including consent withdrawal, please see our Privacy Policy

Reviewer #1: No
